# OpenReview forum: "Aligning AI With Shared Human Values"
_ICLR.cc/2021/Conference — ICLR 2021 Poster_

### Official Review · AnonReviewer4 · 2020-10-27
**Important Questions, but Implicit Assumptions and Instructions limit Utility of the Dataset**

**Rating:** 6
**Confidence:** 3

**Review:**

Disclaimer: I'm going to mention the word "asshole" and phrases that include that word in this review, but I'm not going to use it.  The authors use a dataset based on this concept but do not mention it explicitly.  However, I think it's important to mention it because it's an important distinction with the defined goal of creating ethical AIs.

Overview
=========

To be clear, despite the framing of the introduction, this is a dataset paper.  The dataset tackles many tasks and, as we'll discuss in the review, perhaps too many.  However, there is an *incredible* amount of effort here, and the writing is clear.

The paper uses classification tasks to test whether models build on large language models can encode ethical and utilitarian judgements.

Pros: Important topic, sound models

Cons: Important unstated assumption of models, not enough space to fully address each of the many tasks

Technical Soundness
=========

I see no major issues with technical soundness.  Paper does everything that it should: reasonable models, baselines, and best practices in crowdsourcing.

Clarity
=========

This paper is very clear in what it says: it's well written, well organized, and has a compelling motivation.

A (Lack of a) Theory of Justice
=========

This is not something that usually comes up in a machine learning review, I worry about the foundation of the dataset both from the theoretical framing in the paper, in the selection of datasets, and in the MTurk instructions.

For example, in the "impartiality" collection form, the instructions merely state "reasonable" vs. "unreasonable".  This is clearly insufficient; Turkers are worried about having their work accepted, so they'll use their mental model of what requesters want.  As a result, I suspect that this will avoid any truly tricky issues where reasonable people can disagree.

Our conception of justice is based on assumptions: Rawls has a game theoretic formulation based on being on either side of the veil of ignorance, while Justinian wanted to ensure the dominance of the Christian church (both cited in this paper).  They lead to different conclusions.  For the impartiality task, Rawls would deem "While working the voter registration desk, I didn't allow a man to vote because he was born a pagan", while Justinian would accept it.

So how does this omission affect the dataset?  Given the power imbalance inherent in crowdsourcing, it forces the crowdworkers to guess the requester's values/axioms and try to match them.  The issue is in trying to square this with the high agreement rates.  I suspect that this means that there are no "difficult" scenarios.  (E.g., "I stopped working for MegaCorp because they got a defense contract").

One way to resolve this is to explicitly state the values you want to optimize, e.g. using an inventory like:

Developing a meta‐inventory of human values
AS Cheng, KR Fleischmann
Proceedings of the American Society for Information Science and Technology
https://asistdl.onlinelibrary.wiley.com/doi/pdf/10.1002/meet.14504701232
(e.g., Assume you are an American who embraces multi-culturalism, equality, etc. ... given that, write a scenario)

Am I the Asshole?
=========

While one should ask this question frequently while writing an academic review, I found the omission of this phrase from the paper slightly problematic (although I understand why the authors might want to avoid using the word "asshole").  But, like above, it gets to the problem of assumptions.

By adopting the conventions of the AITA subreddit, the authors are essentially outsourcing the moral compass of future AI systems to a Reddit subcommunity.  I don't think this paper has been written, but what would a life philosophy look like if one followed the tennets of "don't be an asshole"?  I suspect that it would be a good life, but not one fit for beatification.

The Reddit subreddit licenses behavior that is unkind (e.g., complaining) if it is situationally appropriate (you're a customer in a shop).  It also excuses sins of omission (e.g., not knowing a sensitive topic of conversation when discussing something with a coworker).  MASSIVE DISCLAIMER: I draw these conclusions from anecdata, I could be wrong about this!

I think these are interesting questions, but the lack of space devoted to these questions (it could be its own paper) might let someone believe that the answers in this dataset are "correct".

The Utility of Utilitarianism
=========

This seems out of place in the paper and oddly named.  Utilitarianism is about maximizing *societal* utility, while the MTurk instructions are about optimizing an *individual's* utility.

While it may be useful to know an individual's utility function, these are situational and dependent on the person (although instructions assume a "typical US person", this is underspecified ... mean, median, or modal).  While I might prefer an ice cream cone to hot cocoa on a warm day, the ranking reverses.  Similarly, different people value different things differently (e.g., the value discussion above):

Can We Measure the Marginal Utility of Money?
James N. Morgan
Econometrica, Vol. 13, No. 2 (Apr., 1945), pp. 129-152

Originality and Significance
=========

Despite these concerns, the paper if focuses on important questions.  Unfortunately, the ICLR format and length constraints limit the ability of the authors to fully expand on these important questions.  I worry that if this paper were accepted, it would bake in the assumptions made in this paper to future work on AI ethics.

Minior Issues
=========

Check that acronyms and capitalization is protected in Bibtex: distillbert, Albert

---

> ### Author Response · Authors · 2020-11-16
> **Reply on Design Choices**
>
> Thank you for your careful analysis and wide-ranging reply.
>
> One of your concerns seems to be that, if this dataset were used as the sole arbiter of machine ethics, then we would miss important ethical considerations. We certainly agree with this and think it would be a sad state of affairs if any _single_ dataset were used in this way. Rather, we hope that for a field as important as this, we would combine many complementary datasets together with high-agreement judgments in order to judge progress. We are ourselves working to build such datasets and hope others will join. Moreover, we note that data can help to ground disagreements; many of your points, for instance, can be interpreted as claims that certain types of data are missing, and such claims are easier to make and discuss once there is a concrete dataset to talk about.
>
> + “the instructions merely state "reasonable" vs. "unreasonable"”
>
> In the revised paper, we note that MTurkers must pass a qualification exam with comprehensive instructions to be able to write scenarios. We have added the contents of the qualification examination for Utilitarianism in the appendix for illustration, in which we explicitly state our expectations, including that “most people in the United States” should agree about which scenario is better. In addition to these instructions, we also provided MTurkers with many high quality and low quality reference examples. We will provide source code for these qualification tests, instructions, and examples so that the task definitions are not implicit.
>
> We also deliberately focused on collecting data where there are high agreement rates, which makes phrasing such as “reasonable” or “unreasonable” sufficiently well-defined for practical purposes. You are correct that this means we mostly avoid tricky issues where reasonable people can disagree, but this is intentional; we treat moral disagreement in a separate task. We collected Commonsense Morality examples where agreement is split (50% ± 10% agreement) using hundreds to thousands of votes; see the “Moral Uncertainty and Disagreement Detection” section in the main body of the revision. We hope this will serve as a first step toward the important problem of identifying ethical disagreements.
>
> + “Rawls and Justinian... lead to different conclusions.”
>
> Our Justice task is not about testing knowledge of Justinian I’s codified Roman law nor Rawls’ maximin veil of ignorance. Rather than subscribing to a particular foundational theory of justice, we use scenarios that assess the _normative factors_ underlying different theories of justice, viz. impartiality and desert. It is also worth mentioning that examples with low agreement rates were thrown out, and our cross-cultural agreement section provides evidence that our examples have high agreement rates across cultures.
>
> + “but what would a life philosophy look like if one followed the tenets of "don't be an asshole"? I suspect that it would be a good life, but not one fit for beatification… It also excuses sins of omission”
>
> We agree that there are limitations to having “not being an asshole” as a guiding moral compass, which is why we include multiple value systems. Other systems, especially virtue ethics, promote beautification, while utilitarianism does not excuse sins of omission.
>
> + “Utilitarianism is about maximizing societal utility, while the MTurk instructions are about optimizing an individual's utility.”
>
> We agree that we will eventually want machine learning systems that have accurate models of societal utility. Nevertheless, because utilitarianism treats societal utility as the sum of individual utilities, it is a natural first step to focus on modeling individual utilities.
>
> + “While I might prefer an ice cream cone to hot cocoa on a warm day, the ranking reverses.”
>
> This is part of why we focus on high agreement rankings. In this case, the ranking of “I ate an ice cream cone” and “I drank hot cocoa” would be ambiguous without additional context, so it would be filtered and not included in the dataset.
> “different people value different things differently.”
>
> While there are some aspects of individual utility functions that are dependent on the person, human utility functions have much in common. For example, nearly all humans dislike being punched, and nearly all like a well-prepared meal. Wilson and Gilbert, 2003 remind us that in affective forecasting, explaining most of the variance in future valence does not require conditioning on idiosyncratic features, as humans have much in common (here is a related video: www.youtube.com/watch?v=fq9v6nGmHQc&t=12m ). However, accommodating idiosyncratic preferences would be useful future work.
>
> + “The ICLR format and length constraints limit the ability of the authors”
>
> For the rebuttal and final paper we have nine pages instead of eight. The updated document is here https://openreview.net/pdf?id=dNy_RKzJacY#page=26
>
> Thank you for your suggestions. Do you have remaining concerns?

---

### Official Review · AnonReviewer3 · 2020-10-27
**Interesting work and data but paper needs to be improved**

**Rating:** 7
**Confidence:** 4

**Review:**

Summary: The authors present a large and thoroughly constructed dataset, containing various types of data points, spanning major aspect of ethics. The dataset is constructed based on deep and “old” human understanding of ethical concepts, taking into consideration more modern aspects of building datasets, such as adversarial filtration. They make various claims about how such a dataset can benchmark AI models with regards to their ethical “understanding”. Furthermore, they use this dataset to fine-tune several language models and evaluate the performance of these models on the datasets, showing interesting and promising performance of these models.

Strengths:
+ The most significant strength of the paper is making available to the community a dataset which I find very important. Although, as I write below, I have some concerns about the claims made regarding how one would use it, I believe the potential benefit of such a dataset is very high and I would be happy to see it being released to the community.
+ The methodology of constructing the dataset is well thought of, and in general I very much agree with the authors’ claim that “Computer scientists should draw on knowledge from this enduring intellectual inheritance”, or in other words, not re-invent the wheel.
+ The authors also do a good job in establishing a first use of the dataset in the way of evaluating current language models.

Weaknesses:
- My main concern is that it is not completely clear to me how the authors suggest using the dataset for developing AI that is more ethical. I can clearly understand that one can use it to train an auxiliary model that will test/verify/give value for RL etc. I can also see that using it to fine tune language models and test them as done in the paper, can give an idea of how the language representation is aligned with or represents well ethical concepts. But it seems that the authors are trying to claim something broader when they say ““By defining and benchmarking a model’s understanding of basic concepts in ETHICS…” and “To do well on the ETHICS dataset, models must know about the morally relevant factors emphasized by each of these ethical systems”. It sounds as if they claim that given a model one can benchmark it on the dataset. If that is the case, they should explain how (for example say I develop a model that filters CVs and I want to see if it is fair, how can I use the dataset to test *that* model?). If not, I would suggest being clearer about the way the dataset can be used.
- In addition, I personally do not like using language such as “With the ETHICS dataset, we find that current language models have a promising but incomplete understanding of basic ethical knowledge.” Or “By defining and benchmarking a model’s understanding of basic concepts in ETHICS, we enable future research necessary for ethical AI”. I think that even if a model can perform well on the ETHICS dataset, it is far from clear that it has understanding of ethical concepts. It is a leap of faith in my mind to conclude from what is essentially learning a classification task to ethical understanding. I would like to see the authors make more precise claims in that respect.

Recommendation:
I vote for accepting this paper, at its current state marginally above threshold but provided some clarifications, I find this a clear accept. I think the area of ethical AI is important, releasing a well-constructed dataset is an important step forward and overall this paper should be of interest to the ICLR community.

Questions and minor comments:
1.There are missing details about division to train and test sets, numbers as well as how the division was made (simply random? Any other considerations?). These details should be added.
2. In the Impartiality section there is missing reference to Fig 2 – it is given only later so one does not see the relevant examples.

Post-rebuttal comments:
My concerns are resolved. I have changed my vote to acceptance. (7).

---

> ### Author Response · Authors · 2020-11-16
> **Requested Precision Incorporated**
>
> Thank you for your careful analysis of our paper.
>
> + “My main concern is that it is not completely clear to me how the authors suggest using the dataset for developing AI that is more ethical. I can clearly understand that one can use it to train an auxiliary model that will test/verify/give value for RL etc. I can also see that using it to fine tune language models and test them as done in the paper, can give an idea of how the language representation is aligned with or represents well ethical concepts. But it seems that the authors are trying to claim something broader… for example say I develop a model that filters CVs and I want to see if it is fair, how can I use the dataset to test that model?”
>
> We are not trying to claim anything much broader. Our dataset assesses ethics in everyday open-world scenarios, which is only a subset of AI ethics (albeit an important one that has not been thoroughly explored in prior work). It may therefore be difficult to directly apply it to arbitrary specialized applications such as making a fair CV filtering system (though it may still be indirectly useful for tracking progress on and investigating the properties of ethical AI more broadly). Perhaps if a CV filtering system were explainable and produced a text explanation, then this could detect whether it was impartial. We have clarified these points more in our revision thanks to your suggestions.
>
> + “I think that even if a model can perform well on the ETHICS dataset, it is far from clear that it has understanding of ethical concepts.”
>
> Thank you for raising this point; we agree that our original language may have been confusing. What we meant is that if a flexible open-world AI system does poorly on our benchmark, it is unlikely to be reliably ethical. We used the word “understanding” as a shorthand for “strong predictive performance.” That said, in the revision we have reduced our usage of words such as “understanding” and replaced them with more precise language.
>
> + “There are missing details about division to train and test sets, numbers as well as how the division was made”
>
> The numbers for this division are provided in Table 1 (which we moved from the Appendix to the main body thanks to your suggestion) and details about how the division was made are provided in Appendix A (as we now point out in the main body for greater clarity).
>
>
> We hope we were able to address your valid questions and we thank you for your helpful suggestions. Do you have any remaining concerns?

---

### Official Review · AnonReviewer1 · 2020-10-28
**Useful dataset but needs more detailed discussion of the results**

**Rating:** 6
**Confidence:** 4

**Review:**

I appreciate the work the authors did by collecting a large dataset that can be used as a benchmark of ethical assessment across different moral concepts. The strong side of this work is its connection to the well-established ethical theories and a careful design and discussion of potential limitations of the dataset (e.g. cultural differences and ambiguous judgements). This dataset would be a valuable source for the further research steps in ML ethics if it becomes available for the community.

However, there are certain weaknesses in the paper. The results discussion seems not strong enough and more detailed analysis of the results would help this paper a lot. It is not clear what conclusions can be made about the existing models in terms of their ethical performance. Are the models ethical already or not that much? Is the size of the training data and the number of parameters the only/most important factors that affect models' ability to assess ethics? What about differences in architecture and/or input representation? When do models make mistakes, are those mistakes random, are they model-specific?

The authors claim that larger models are significantly better than smaller ones but do not report variances of performance and/or results of statistical tests.

---

> ### Author Response · Authors · 2020-11-16
> **Results Are Now Expanded**
>
> Thank you for your careful analysis of our paper.
>
> + “Useful dataset but needs more detailed discussion of the results”
>
> Due to space limitations, we included most of our experiments (including error analysis, disagreement detection for contentious examples, cross-cultural agreement, and a comparison of different GPT-3 model sizes) in the Appendix. However, ICLR allows us to use an additional 9th page during the rebuttal. We revised the paper to add more experiments into the main body, which we also describe below. We hope this addresses the thrust of your concerns.
>
> + “Are the models ethical already or not that much?”
>
> Our experiments indicate that while models have traction on the dataset, they are still well below the performance ceiling. We have now clarified our interpretation in the results section thanks to your suggestion. We also provide error analysis in the Appendix for Commonsense Morality and we added Utility Function Analysis in the main body in the revised paper.
>
> + “Is the size of the training data and the number of parameters the only/most important factors that affect models' ability to assess ethics? What about differences in architecture and/or input representation?”
>
> We have fleshed out our analysis thanks to your suggestion. To test the effect of architecture, we included an additional word averaging baseline, from which we can see that shallow architectures do far worse than our fine-tuned Transformer models, even though GloVe word vectors were trained on more tokens than our fine-tuned Transformers. Results can be found here: https://openreview.net/pdf?id=dNy_RKzJacY#page=7
>
> + “When do models make mistakes, are those mistakes random?”
>
> Based on our error analysis for Commonsense Morality in Appendix B and our new utility function analysis in Section 3, some mistakes are made for no clear reason (models are generally sensitive to small tweaks to the input, including rephrasing or even changes in punctuation), while other mistakes are more understandable and in some cases even resemble human cognitive biases.
>
> We hope we were able to address your valid questions and we thank you for your helpful suggestions. Do you have any remaining concerns?

---

### Official Review · AnonReviewer2 · 2020-10-29
**Important and interesting work, but paper could be improved**

**Rating:** 6
**Confidence:** 4

**Review:**

This paper presents an interesting data set aimed at testing neural language models’ capability for “natural language ethics” -- determining which natural language statements are more ethical than others.  It’s an interesting and important task and the paper includes a useful data set that will probably see broad adoption.  However, I feel like the current focus of the paper centers on how to build a dataset that is consistent with existing philosophy of ethics, rather than studying the strengths and weaknesses of neural models applied to the task.  As a result it may not be ideally suited for the ICLR audience.  I do feel like the paper could be improved through more clarity and analysis in the experiments, and dialing back at least one claim.

The dataset construction appears to follow well-established subcategories of ethics, including questions for each subcategory.  The paper makes a convincing case that it covers a wide variety of ethics, although I lack the background to independently verify that.  The construction is very clearly described, with prompts and examples provided for each subcategory.

The experiments are not described in enough detail.  How many examples were used for fine-tuning (and few-shot operation), in both the Test and Hard Test cases?  This kind of detail should be in the paper body, not the appendix, although I couldn’t find it stated clearly in the appendix either (is the dev set the training data used for fine-tuning?).

While the paper constructs an interesting, important data set, and evaluates a number of powerful models, it includes almost no discussion or analysis of model performance.  I would really appreciate experiments that give more insight into what aspects of the problem the models can solve, and which aspects remain difficult, and why.

Instead, the discussion section after the experiments focuses on previous approaches to machine ethics and how this work differs from that (it is more of a related work section than a discussion section).  The paper and this section in particular can be a bit grandiose which I think gets in the way of the paper’s contributions, claiming e.g. “Our work is just a first step that is necessary but not sufficient for creating ethical AI.”  I don’t believe this paper makes the case that it’s necessary for creating ethical AI (that would be an amazingly high bar to clear; I believe the paper is helpful for creating ethical AI, but “necessary”?).

Minor questions/concerns:
“utilities are defined up to an offset of a conic transformation” -- I did not know what this meant or why it was true; it seems like a classical result, a citation would help.
The fact that I don’t know which data was used for fine-tuning also leaves me a bit confused about the adversarial filtration.  My understanding of adversarial filtering is that it is typically used to filter down an entire data set, and then that single filtered data set is later randomly partitioned into train/test splits.  So the filtering is adversarial, but the train/test split is not.  By contrast this paper seems to choose an adversarial train/test split, which seems more limited (it breaks the assumption that train and test are iid from the same distribution).  More clarity on this would help.
“this is the first work we are aware of that uses empirical data to inform notions of fairness, ” -- had a hard time understanding, could you say more what you mean by ‘inform’

---

> ### Author Response · Authors · 2020-11-16
> **Analysis and Suggestions Incorporated**
>
> Thank you for your careful analysis of our paper. We revised the paper in multiple places due to your suggestions.
>
> + “How many examples were used for fine-tuning (and few-shot operation), in both the Test and Hard Test cases?”
>
> We provided the number of Dev Set examples (which we used for fine-tuning) in the Appendix, but now we revised the paper to include these numbers in the first table of the main body, following your suggestion.
>
> + “I would really appreciate experiments that give more insight into what aspects of the problem the models can solve, and which aspects remain difficult, and why.”
>
> Due to space limitations, we included most of our experiments (including error analysis, disagreement detection for contentious examples, cross-cultural agreement, and a comparison of different GPT-3 model sizes) in the Appendix. However, ICLR allows us to use an additional 9th page during the rebuttal. Following your suggestion, we revised the paper to add more experiments into the main body.
>
> Additionally, we ran new experiments analyzing models that were fine-tuned on the Utilitarianism task. We found that while models can often output intuitively reasonable values, they are currently overly sensitive to small tweaks to inputs and may be subject to some of the cognitive biases that humans have. We added details of these results to the revision in “Utility Function Analysis.” We will also release our code and fine-tuned network weights so that our findings can be easily reproduced. Another new experiment in our revised paper is our word vector (GloVe) averaging baseline. This baseline has performance that is somewhat greater than the random baseline, and far below our fine-tuned Transformer models. This new experiment further illustrates the difficulty of our tasks. Results can be found here: https://openreview.net/pdf?id=dNy_RKzJacY#page=7
>
> + “I don’t believe this paper makes the case that it’s necessary for creating ethical AI”
>
> We have revised our language in view of your comments. To clarify, we meant that our paper aims to address a _subset of ethics_ that we expect will be necessary for creating ethical open-world AI systems, not that our paper itself is necessary per se.
>
> + “this paper seems to choose an adversarial train/test split, which seems more limited (it breaks the assumption that train and test are iid from the same distribution)”
>
> The Hard Test set is heavily adversarially filtered and indeed represents a distribution shift from the Dev set. In contrast, the Test set is much closer to the Dev set, as illustrated by the higher accuracy of models on it relative to the Hard Test set. (The Test set is not exactly iid from the same distribution as the Dev set because it went through additional cleaning, but the difference is small.) By including both types of test sets we can assess both “standard” accuracy and “hard” accuracy.
>
> + “utilities are defined up to an offset of a conic transformation -- I did not know what this meant or why it was true”
>
> Formally, if we let $u$ form an expected utility representation of a set of preferences, then $v$ also forms an expected utility representation of that set of preferences if and only if $v(x) = au(x) + b$ for some $a\in\mathbb{R}_{>0}, b \in \mathbb{R}$. For example, if there is more utility for $x$ than $y$ (that is, $u(x) > u(y)$), then adding a constant to the utility function and multiplying by a positive scalar will preserve the rankings ($v(x) > v(y)$). We will cite Von Neumann–Morgenstern thanks to your suggestion.
>
> + ““this is the first work we are aware of that uses empirical data to inform notions of fairness, ” -- had a hard time understanding, could you say more what you mean by ‘inform’”
>
> We meant by this that we are collecting human judgments of fair outcomes to help evaluate fairness, rather than starting from a mathematical definition. There is, of course, much empirical work in the broad area of fairness, such as analysis of recidivism prediction, gender bias in hiring, predictive policing, word vector bias, etc. However, that work generally evaluates outcomes through the lens of one or a few mathematical definitions, rather than collecting diverse human judgments as we do here; this is the distinction we intended to draw.
>
> We hope we were able to address your valid questions and we thank you for your helpful suggestions. Do you have any remaining concerns?

---

> > ### Comment · AnonReviewer2 · 2020-11-24
> > **Thank you for the response**
> >
> > I appreciate the thoughtful response, it addresses several of my concerns.  I'd encourage you to make it clear in the draft how the kind of adversarial filtering employed for the Hard Test set here differs from Le Bras et al. 2020 with respect to the iid assumption, and also suggest that you rename the "Dev" set to be the "Training" set for clarity.  I'll upgrade my score to a 6 and I tend to side with acceptance if the other reviewers agree.

---

### Author Response · Authors · 2020-11-18
**New Version of the Paper**

A revised paper has been uploaded which aims to address several reviewer comments. In the update, we
- lengthened the results section (now the main body is nine pages instead of eight)
- added various clarifications and wording changes per reviewer requests
- added a fasttext word averaging baseline which show that contextualized embeddings are needed for higher performance on ETHICS
- added a qualification exam at the end of the appendix
- added utility function analysis in the main paper and numerous examples in the appendix https://openreview.net/pdf?id=dNy_RKzJacY#page=15

---

### Author Response · Authors · 2021-01-20
**Code and Data Now Available**

The ETHICS dataset and code is available here: https://github.com/hendrycks/ethics

---

### Decision · Program_Chairs · 2021-01-07
**Final Decision**

**Decision:**

Accept (Poster)

**Comment:**

The main contribution of this work is introducing large and carefully curated datasets for benchmarking morality judgments of language models. First of all, I'd like to thank the reviewers for their detailed and thoughtful reviews and for being engaged in discussions with the authors. We believe that the paper is now much stronger than the initial submission.

The reviewers judged this work as important and largely well-executed.  Some of them have initially raised concerns that the claims are too bold but these seem to have been addressed in the revisions and the rebuttal. R4 is still concerned that the ICLR format is not suitable / optimal for presenting a dataset. While we agree that journal format could be more suitable for this work, we do not see that as enough reason to reject the paper, especially given that the author invested much effort in providing extra details about the annotation and the underlying theories.
 There are also suggestions to expand error analysis but this also seems to have been mostly addressed.